SciPost Physics

Submission

# Correlations of quantum curvature and variance of Chern numbers

Omri Gat[1*], and Michael Wilkinson[2,3†]

**1** Racah institute of Physics, Hebrew University, Jerusalem 91904, Israel
**2** Chan Zuckerberg Biohub, 499 Illinois Street, San Francisco, CA 94158, USA
**3** School of Mathematics and Statistics, The Open University, Walton Hall, Milton Keynes, MK7 6AA, England
*omrigat@mail.huji.ac.il [†] michael.wilkinson@czbiohub.org

December 7, 2020

## Abstract

We analyse the correlation function of the quantum curvature in complex quantum systems, using a random matrix model to provide an exemplar of a universal correlation function. We show that the correlation function diverges as the inverse of the distance at small separations. We also define and analyse a correlation function of mixed states, showing that it is finite but singular at small separations. A scaling hypothesis on a universal form for both types of correlations is supported by Monte-Carlo simulations. We relate the correlation function of the curvature to the variance of Chern integers which can describe quantised Hall conductance.

# 1 Introduction

The *quantum curvature* $\Omega_n$ of an eigenstate of a quantum system (with index $n$) is an object which characterises the sensitivity of the eigenfunction to variation of parameters of the Hamiltonian. It plays an important role in the the dynamics of quantum systems [1–4]. In this paper we characterise fluctuations of the quantum curvature in generic *complex quantum systems* (which have many energy levels and no constants of motion or Anderson localisation effects). We analyse correlations of the quantum curvature in parameter space using random matrix models [5, 6], which are applicable to generic complex quantum systems upon application of a scaling transformation. We relate the correlation function to statistics of the Chern numbers, which arise in the analysis of quantised conductance phenomena.

The quantum curvature is defined for a system with a Hamiltonian $\hat{H}$, which depends upon at least two parameters (with the position in parameter space being denoted by $\mathbf{X} = (X_1, X_2)$). It may be defined for a non-degenerate level by writing

$$\Omega_n \, \mathrm{d}X_1 \wedge \mathrm{d}X_2 = -\mathrm{i} \, \mathrm{tr} \left[ \hat{P}_n \mathrm{d}\hat{P}_n \wedge \mathrm{d}\hat{P}_n \right] \ , \tag{1}$$

where $\hat{P}_n = |\phi_n\rangle\langle\phi_n|$ is the projection onto the eigenstate $|\phi_n\rangle$ of $\hat{H}$ with index $n$. Several dynamical applications of $\Omega_n$ have been discovered. Mead and Truhlar [1] showed that when $\mathbf{X}$ is varied slowly, there is a component of the Born-Oppenheimer reaction force which is proportional to the product of $\Omega_n$ and to the rate of change of parameters, $\dot{\mathbf{X}}$. Related applications arise in solid-state physics [2, 3]. Berry [4] emphasised that the integral of $\Omega_n$ over an arbitrary surface is proportional to a 'geometric phase' which appears in adiabatic approximations to the wavefunction, and this is our motivation for referring to $\Omega_n$ as a 'quantum curvature'. Note, however, that $\Omega_n$ is identically zero if the Hamiltonian can be represented by a real-valued matrix.

In the applications considered in [1,3,4], the parameter $\mathbf{X}$ is varied slowly as a function of time. This can result in transitions between energy levels, so that the system will evolve to a mixed state. In particular, near-degeneracies of levels will allow Landau-Zener transitions between states [7], which results in a diffusive spread of the probability of a given level being occupied [8]. For this reason we shall also consider a 'smoothed' curvature, $\bar{\Omega}_\varepsilon(E)$, involving a weighted average of $\Omega_n$ over an energy interval of length $\varepsilon$ centred at $E$:

$$\bar{\Omega}_\varepsilon(E, \mathbf{X}) = \sum_n \Omega_n(\mathbf{X}) w_\varepsilon(E - E_n(\mathbf{X}))$$

$$w_\varepsilon(E) = \frac{1}{\sqrt{2\pi}\varepsilon} \exp(-E^2/2\varepsilon^2) \ . \tag{2}$$

A Gaussian smoothing is preferred here because this is the kernel for diffusive spread over energy levels. We assume $\rho\varepsilon \gg 1$, so that many levels are included in the average, but that $\varepsilon$ is small compared to other energy scales in the system. Another motivation for considering $\bar{\Omega}_\varepsilon$ is that we shall see that the dependence of its statistics upon $\varepsilon$ allows inference about correlation of the $\Omega_n$ between different values of the level index, $n$.

It is known that quantum systems with many energy levels may exhibit universal behaviour if there are no constants of motion other than the Hamiltonian, and no Anderson

localisation effects . These universal properties are most conveniently computed using random matrix ensembles [5,6,9]. We shall discuss the use random matrix models in section 2. There we review how random matrix approaches have been extended to systems where the Hamiltonian depends smoothly on a parameter [10–15], and introduce (section 2.4) a hypothesis on the universal form of the correlation functions of the quantum curvature. In order to compute this universal correlation function, we consider a random matrix model in which the Hamiltonian depends smoothly upon two parameters. It suffices to consider a model (introduced in section 2.5) in which $\langle \Omega \rangle = 0$, and for which the statistics are homogeneous and isotropic in parameter space. For this model we investigate two correlation functions

$$
\begin{aligned}
C_{nm}(X) &= \langle \Omega_n(\mathbf{X}, 0)\Omega_m(0, 0)\rangle \ , \\
\mathcal{C}(\Delta E, X) &= \langle \bar{\Omega}_\varepsilon(E_0 + \Delta E, \mathbf{X})\bar{\Omega}_\varepsilon(E_0, 0)\rangle
\end{aligned}
\tag{3}
$$

where $X = |\mathbf{X}|$ (angle brackets denote expectation values throughout).

Thouless *et al.* [2] showed that $\Omega_n$ arises in an evaluation of the Hall conductance via the Kubo formula, and that the the Hall conductance of a filled band is quantised by arguing that

$$
N_n = \frac{1}{2\pi}\int_{\mathrm{BZ}} \mathrm{d}\mathbf{X} \ \Omega_n(\mathbf{X})
\tag{4}
$$

takes integer values (where, in this case, the parameter $\mathbf{X}$ is a Bloch wavevector and where the integral runs over the Brillouin zone). This topological invariant, known as the Chern index [16]. The integral of $\Omega_n/2\pi$ over any closed two-dimensional manifold is also an integer-valued topological invariant. Later Thouless extended these results to show quantised conductance in 'sliding' periodic potentials [3], using adiabatic approximations, akin to those in [1], rather than the Kubo formula. We shall use our results on the correlation function $C(X)$ to compute the variance $\mathrm{Var}(N_n)$ of the Chern integers in our model. We also argue that the correlation function of the Chern integers in complex quantum systems is well-approximated by:

$$
\langle N_n N_m\rangle - \langle N_n\rangle\langle N_n\rangle = \frac{1}{2}\mathrm{Var}(N_n)\left[2\delta_{nm} - \delta_{n,m+1} - \delta_{n,m-1}\right]
\tag{5}
$$

(we have $\langle N_n\rangle = 0$ for our random matrix model). The Chern integers can change by $\pm 1$ when energy bands become degenerate [17], and equation (5) is consistent with the effects of these degeneracies being uncorrelated between different levels.

While the general question of spectral statistics of systems depending on parameters has been quite extensively studied, relatively little attention has been devoted specifically to the statistical properties of the quantum curvature. In an early paper, Berry and Robbins [18] studied semiclassical approximations for the curvature in systems with a chaotic classical limit using Gutzwiller's periodic orbit theory [19]. The expression for the quantum curvature obtained in [18] is not rigorously defined, and while it has been successfully applied to families of unitarily equivalent Hamiltonian [20], the semiclassical curvature statistics of generic families is still unknown. While not dealing directly with the curvature, Walker and Wilkinson [21] studied the related questions of the statistics of degeneracies, where the curvature diverges, and Chern numbers in random matrix fields, arguing that they are universal, and developing a scaling theory for them. Berry and Shukla [22–24] studied the single-point probability density function $p(\Omega)$ of the curvature, and showed that the distribution has a power law decay $p(\Omega) \sim |\Omega|^{-5/2}$ for $|\Omega|$ large. The tails of the curvature distribution are dominated by near-degeneracy events, and the decay exponent, determined by the codimension of the degeneracies, is small enough that the

variance of the single-level curvature $\langle \Omega^2 \rangle$ is infinite, while the expectation value $\langle \Omega \rangle = 0$ converges due to symmetry.

As a consequence of the broad distribution of $\Omega_n$, the single level correlation functions $C_{nm}(X)$, $m = n, n \pm 1$, which are finite for $X \neq 0$, diverge as $X \to 0$. The smoothed curvature correlation function $\mathcal{C}_\varepsilon(\Delta E, X)$ is finite for all $X$, but fluctuations due to near degeneracies make it singular at short separations with a discontinuous derivative at $X = 0$. We calculate the contribution of near-degeneracy fluctuations to the two-point correlation functions, which together with the one-point correlation function of the smoothed curvature completely determines the short-separation behaviour of both the single-level and the smoothed curvature correlation functions. This is the first main theoretical result of this paper; the other main result is the scaling forms of the two types of curvature correlation function that are conjectured to be universal. Both the short-distance and the scaling of the correlations are compared with comprehensive Monte-Carlo simulations, that support the theoretical prediction in the large-matrix-size limit.

In section 2 we describe and motivate the random matrix models that we use. Section 3 discusses our theoretical and numerical results on the correlation functions of the single-level curvature $\Omega_n$. The analogous discussion for the correlation function of the smoothed curvature $\bar{\Omega}_\varepsilon$ is the subject of section 4. We consider the implications for Chern numbers in section 5, estimating their variance and presenting an argument in support of equation (5). Finally, section 6 discusses our conclusions and prospects for further studies.

## 2 Random matrix model

There is ample evidence for universality of the properties of *complex quantum systems* (loosely defined as systems with many energy levels, which do not have Anderson localisation effects or constants of motion which are independent of the Hamiltonian) [5,9]. The universal properties are manifest in spectral properties which involve small energy scales, or equivalently in dynamical behaviour on long timescales. Hermitean random matrix models of complex quantum systems, and have the attractive feature that they may be used to compute the universal properties analytically [6].

Consider a Hamiltonian depending upon two parameters, $X_1$, and $X_2$ (write $\mathbf{X} = (X_1, X_2)$). The quantum curvature $\Omega_n$ is a fundamental characterisation of the sensitivity to parameters of the projection $\hat{P}_n$ onto the level with index $n$. Following [4], we can use perturbation theory to express $\Omega_n$ in terms of matrix elements of derivatives of the Hamiltonian, and energy levels. This leads to the expression

$$
\begin{aligned}
\Omega_n &= \mathrm{Im} \sum_{m \neq n} \frac{\partial_1 H_{nm} \partial_2 H_{mn} - \partial_2 H_{nm} \partial_1 H_{mn}}{(E_n - E_m)^2} \\
&= -\mathrm{i} \sum_{m \neq n} \frac{\partial_1 H_{nm} \partial_2 H_{mn} - \partial_2 H_{nm} \partial_1 H_{mn}}{(E_n - E_m)^2} \ .
\end{aligned}
\tag{6}
$$

Here $E_n$ are eigenvalues of the Hamiltonian $\hat{H}(X_1, X_2)$ with eigenvectors $|\phi_n\rangle$ and $\partial_i H_{nm}$ are matrix elements of derivatives of the Hamiltonian in its eigenbasis:

$$
\partial_i H_{nm} = \langle \phi_n | \frac{\partial \hat{H}}{\partial X_i} | \phi_m \rangle \ .
\tag{7}
$$

Equation (6) will be the basis for our calculations of the statistics of the curvatures, $\Omega_n$. In order to evaluate (6) we require information about statistics of both energy levels and matrix elements.

## 2.1 Distribution of energy levels

The statistics of the energy levels $E_n$ for complex quantum systems have been very extensively studied [5, 6, 9]. It is hypothesised that short-ranged statistical properties of the spectrum, such as the probability distribution of the spacing of adjacent levels, are universal once the energy levels are transformed to levels with unit mean spacing. If $N(E)$ is a smooth function representing the mean integrated density of states, the transformed levels are $e_n = N(E_n)$. In many cases the complex system is close to a classical limit, and the integrated density of states can be derived from the Weyl rule [19]. There are three universality classes of bulk level statistics, which are exemplified by three Gaussian random matrix ensembles. Individual matrices of our model have the Gaussian unitary ensemble (GUE) statistics, because the curvature is odd under time reversal, and therefore zero in the other Gaussian ensembles (orthogonal and symplectic) that obey time-reversal symmetry. Equation (6) shows that $\Omega_n$ diverges if $E_n$ approaches degeneracy with either the level above or below. For this reason the probability distribution function of the separation of two levels will play a central role in our analysis. If $\rho(E) = \mathrm{d}N/\mathrm{d}E$ is the mean density of states, then the PDF of the normalised separation $S = (E_{n+1} - E_n)\rho$ is well approximated by the Wigner surmise: for the GUE this takes the form

$$P(S) = \frac{32}{\pi^2} S^2 \exp(-4S^2/\pi) \ . \tag{8}$$

The exact form of the distribution is complicated but when the matrix size is large it tends to a universal limit, which for $S \ll 1$ has the asymptotic approximation

$$P(S) \sim \frac{\pi^2}{3} S^2 \ . \tag{9}$$

## 2.2 Distribution of matrix elements

In order to compute the statistics of the $\Omega_n$, we also need information about the statistics of the matrix elements of derivatives of the Hamiltonian with respect to its parameters. In complex quantum systems, theoretical arguments and numerical experiments [10] support the use of a model where the off-diagonal matrix elements (7) are statistically independent of each other, independent of the energy levels, and approximately Gaussian distributed, with mean value equal to zero. To complete the characterisation of the distribution of these elements, we must specify their variance. The variance is a function of the energies of the two states, and we define

$$
\sigma_{ij}^2(E, \Delta E) = \frac{1}{\rho(E + \Delta E/2)\rho(E - \Delta E/2)} \\
\times \sum_n \sum_m \partial_i H_{nm} \partial_j H_{mn} w_\varepsilon(E - (E_n + E_m)/2) w_\varepsilon(\Delta E - (E_n - E_m)) \tag{10}
$$

(where the energy window function $w_\varepsilon$ is used instead of a Dirac delta function, so that $\sigma_{ij}^2$ has a smooth dependence upon its arguments). If the complex quantum system has a good classical limit, the covariance $\sigma_{ij}^2(E, \Delta E)$ can be calculated using the method described in [25]. Because (6) implies that small energy separations dominate the sum, it is the value of $\sigma_{ij}^2(E, \Delta E)$ with $\Delta E \to 0$ that determines the statistics of the curvatures $\Omega_n$. We can always make a locally linear transformation of the coordinates $(X_1, X_2)$ so that the covariance $\sigma_{ij}^2$ is a multiple of the identity, with diagonal elements denoted by $\sigma^2$. For convenience, the universal form for the correlation functions that we consider in this work will be computed in such an isotropic coordinate system. However, for the purposes

of understanding the dimensions of expressions it is convenient to distinguish between derivatives with respect to $X_1$ and $X_2$. For this reason we shall express the statistics of $\Omega_n$ in terms of two variances

$$\sigma_i^2 = \langle |\partial_i H_{n+1,n}|^2 \rangle . \tag{11}$$

where the angular brackets indicate an average over $n$: in terms of equation (10), we identify $\sigma_i^2 = \sigma_{ii}^2(E, 0)$.

## 2.3 Projection into a two-level subspace

In the case where two levels become nearly degenerate, we can approximate $\Omega_n$ by a projection onto a two-level subspace: in section 3 this approach will be used to determine the behaviour of $C(X)$ analytically in the limit $X \to 0$. Write

$$\hat{H}(\mathbf{X}) = \hat{H}_0 + \sum_{i=1,2} \frac{\partial \hat{H}}{\partial X_i} X_i \tag{12}$$

and the matrix elements are

$$H_{nm} = E_n \delta_{nm} + \sum_{i=1,2} \partial_i H_{nm} X_i \tag{13}$$

(where the states $|\phi_n\rangle$ are eigenstates at $\mathbf{X} = \mathbf{0}$). Assume that the levels $n$, $n+1$ are nearly degenerate at $\mathbf{X} = \mathbf{0}$, with the separation $E_{n+1} - E_n$ being much smaller than other gaps in the spectrum. In this case the curvature close to $\mathbf{X} = \mathbf{0}$ is determined by the projection of the Hamiltonian into the two-level subspace spanned by $|\phi_n\rangle$ and $|\phi_{n+1}\rangle$. The projection of the Hamiltonian into this subspace is represented by a $2 \times 2$ matrix, which can be written in the form

$$\tilde{H}(X_1, X_2) = \sum_{i=0}^{3} h_i(X, Y) \tau_i \tag{14}$$

where the $\tilde{\sigma}_i$ are Pauli matrices ,

$$\tau_1 = \begin{pmatrix} 0 & 1 \\ 1 & 0 \end{pmatrix} , \quad \tau_2 = \begin{pmatrix} 0 & -\mathrm{i} \\ \mathrm{i} & 0 \end{pmatrix} , \quad \tau_3 = \begin{pmatrix} 1 & 0 \\ 0 & -1 \end{pmatrix} \tag{15}$$

with $\tau_0$ equal to the $2 \times 2$ identity matrix. Because adding multiples of the identity does not change the eigenvectors (and therefore leaves the curvature invariant), we assume without loss of generality that $h_0 = 0$. Close to the origin the projected Hamiltonian is then

$$\tilde{H} = \epsilon \tau_3 + \sum_{i=1}^{3} \sum_{j=1,2} W_{ij} \tau_i X_j \tag{16}$$

Here

$$\epsilon = \frac{E_{n+1} - E_n}{2} , \quad W_{ij} = \left. \frac{\partial h_i}{\partial X_j} \right|_{X_1 = X_2 = 0} . \tag{17}$$

The $W_{ij}$ are related to the matrix elements of the derivatives as follows:

$$W_{1,j} = \mathrm{Re}[\partial_j H_{n+1,n}] , \quad W_{2,j} = \mathrm{Im}[\partial_j H_{n+1,n}] , \quad W_{3,j} = \frac{\partial_j H_{n+1,n+1} - \partial_j H_{n,n}}{2} . \tag{18}$$

In a complex system, the matrix elements $\partial_j H_{nm}$ appear random. For a system with a complex Hermitean Hamiltonian, we expect that $\mathrm{Re}\,\partial_i H_{n+1,n}$, $\mathrm{Im}\,\partial_i H_{n+1,n}$ are independent Gaussian variables, with mean equal to zero and variance $\sigma_i^2/2$. The diagonal matrix

elements need not have a mean value equal to zero (as evidenced, for example, by semi-classical calculations on chaotic quantum systems, presented in [14, 26]). However, using arguments about unitary invariance of the ensemble of Hamiltonians, it is argued that the variance of the diagonal elements is $\text{Var}[\partial_i H_{n+1,n+1}] = \text{Var}[\partial_i H_{n,n}] = \sigma_i^2$ [10, 26]. Because these elements are independent, $W_{3,i} = [\partial_i H_{n=1,n+1} - \partial_i H_{n,n}]/2$ has variance $\sigma_i^2/2$. We conclude that the $W_{i,j}$ are Gaussian random variables with mean value zero and variance

$$\langle W_{i,j}^2 \rangle = \frac{\sigma_j^2}{2} \ . \tag{19}$$

## 2.4 Universality hypothesis for curvature correlation

The universality hypothesis is most extensively supported for energy level statistics [5, 9], but there is also strong evidence that it holds for parametric dependence of energy levels [10–15], and by extension it should also hold for dynamical properties [8].

In the case of a system which depends upon a single parameter $X$, it is argued [10] that the eigenfunctions depend very sensitively upon parameters, so that correlation functions decay on a characteristic length scale $\Delta X$ upon which the eigenfunction lose their identity. Furthermore, perturbation theory indicates that $\langle \phi_n | \partial \hat{H}/\partial X | \phi_{n+1} \rangle \Delta X \sim \overline{\Delta E}$, where $\overline{\Delta E}$ is the typical separation of energy levels. Because the typical size of the matrix element is $\langle \phi_n | \partial \hat{H}/\partial X | \phi_{n+1} \rangle \sim \sigma$, and the typical spacing of levels is $\overline{\Delta E} \sim \rho^{-1}$, we expect that correlation functions will be functions of the dimensionless variable $\rho\sigma\Delta X$, and this is in accord with numerical investigations [10, 12].

In order to define the quantum curvature, however, we must consider a Hamiltonian which depends upon more than one parameter. Let us assume that our system has two parameters, $\mathbf{Y} = (Y_1, Y_2)$ say, and that the matrix elements of derivatives with respect to the $Y_i$ variables have a covariance $\Sigma_{ij}^2$ (defined by analogy with equation (10)). We can apply a smooth transformation of the parameter space to produce a set of transformed coordinates $\mathbf{X} = (X_1, X_2)$, so that small displacements in parameter space close to $\mathbf{Y}$ are described by a unimodular $2 \times 2$ matrix $\tilde{M}$:

$$\delta\mathbf{X} = \tilde{M} \ \delta\mathbf{Y} \ , \quad \det(\tilde{M}) = 1 \ . \tag{20}$$

We shall calculate the correlation functions in these transformed coordinates, $\mathbf{X} = (X_1, X_2)$. We choose the transformation matrix $\tilde{M}$ so that the covariance matrix $\tilde{\sigma}^2$ (with elements $\sigma_{ij}^2$) is a multiple of the identity matrix, with diagonal elements equal to $\sigma$). If these diagonal elements are denoted by $\sigma^2$, then $\tilde{M}$ satisfies

$$\tilde{\Sigma}^2 = \tilde{M}\tilde{\sigma}^2\tilde{M}^{\mathrm{T}} = \sigma^2\tilde{M}\tilde{M}^{\mathrm{T}} \ , \quad \sigma^4 = \det(\tilde{\Sigma}^2) \ . \tag{21}$$

Now consider the form of the correlation function in the isotropic coordinates, $C_{nn}(X)$, which must be a function of $\sigma_1$, $\sigma_2$ and $\rho$. Dimensional considerations imply that $C_{nn}$ is proportional to $\sigma_1^2\sigma_2^2\rho^4$. In terms of the transformed variables, in which the covariances are diagonal ($\sigma_{ij}^2 = \sigma^2\delta_{ij}$), the correlation function takes the form

$$C(X) = \sigma^4\rho^4 f(\rho\sigma X) \tag{22}$$

where $f(\cdot)$ is a universal function. We shall determine $f(x)$ numerically, and compute its asymptotic behaviour as $x \to 0$ analytically. In the original variables, where the coordinate dependence is not isotropic, we have

$$C(\mathbf{Y}) = \det(\tilde{\Sigma}^2)\rho^4 f\left(\rho[\det(\tilde{\Sigma}^2)]^{1/4}|\tilde{M}\mathbf{Y}|\right) \ . \tag{23}$$

The arguments leading to (22) are immediately applicable to off-diagonal correlation functions $C_{n,n+s}$ with fixed $s$, so that

$$C_{n,n+s}(X) = \sigma^4 \rho^4 f_s(\rho\sigma X) , \tag{24}$$

with a set of universal scaling functions $f_s(x)$.

The smoothed curvature correlation function depends on the energy separation $\Delta E$ in addition to the parameter separation $X$. In section 5 we show that $\mathcal{C}$ is proportional to $\sigma_1^2 \sigma_2^2 \rho^3 / \varepsilon^3$ and argue that its scaling form is

$$\mathcal{C}(\Delta E, X) = \frac{\pi^{3/2}}{6} \frac{\sigma^4 \rho^3}{\varepsilon^3} g(\rho\sigma X, \Delta E/\varepsilon) \tag{25}$$

in the isotropic coordinates (the dimensionless coefficient is chosen so that $g(0,0) = 1$) and calculate explicitly the small-$x$ asymptotics of $g(x,y)$ for any $y$. Furthermore we shall determine $g(x,y)$ numerically for all $x$ and $y$, and confirm that it is indeed universal.

## 2.5  Random matrix fields on the two sphere

We performed our numerical studies on a field of $M \times M$ random matrices taking values on the two-sphere. At each point, the statistics of the matrix field are representative of the Gaussian unitary ensemble (GUE), as defined in [6]. By choosing the distribution that is homogeneous and isotropic, the model is fully specified by $\langle H \rangle = 0$ and the two-point matrix element correlation function

$$\langle H_{ij}(\mathbf{X}) H_{i'j'}^*(\mathbf{X}') \rangle = c(\theta) \delta_{ii'} \delta_{jj'} \tag{26}$$

where $\theta$ is the angle subtended by the points $\mathbf{X}$ and $\mathbf{X}'$ on the sphere; $c$ is a smooth function of $\theta$ with $c(0) = 1$ and $c'(0) = 0$, making the random matrix field realisations smooth functions on the sphere with variance equal to unity.

The simulation results shown below were all obtained for a Gaussian correlation function $c(\theta) = \exp(-\theta^2/2\tilde{\theta}^2)$, where $\tilde{\theta}$ is a parameter of the model. For this model, the covariance coefficients $\sigma_{ij}$ of the matrix element variances form a diagonal matrix, so that the coefficients in equation (11) are $\sigma_1 = \sigma_2 = 1/\tilde{\theta}$. The single point distribution implied by (26) is standard GUE, so that when $M$ is large the mean density of states is well-approximated by Wigner's 'semicircle law' [6],

$$\rho(E) = \frac{\sqrt{4M - E}}{2\pi} , \qquad |E| \leq 2M , \tag{27}$$

and zero otherwise.

# 3  Correlation function of the curvature

## 3.1  Small-separation asymptotics

Consider the form of the correlation function $C(X)$ in the limit as $X \to 0$. In this limit the correlation function diverges, due to near-degeneracies, and we can calculate its form using the projection into a two-dimensional sub-space, as considered in subsection 2.3.

The quantum curvature 2-form, denoted by $\tilde{\Omega}$, is described by a single coefficient $\Omega_n$ when expressed in terms of the coordinates $(X_1, X_2)$:

$$\tilde{\Omega} = \Omega_n \mathrm{d}X_1 \wedge \mathrm{d}X_2 . \tag{28}$$

We can also write $\tilde{\Omega}$ using the coefficients $h_i$ (defined in equation (16)) as coordinates, in which case it is expressed in terms of components $\Omega_{jk}$,

$$
\begin{aligned}
\tilde{\Omega} &= \sum_{\substack{j,k \\ j<k}} \Omega_{jk} \, \mathrm{d}h_j \wedge \mathrm{d}h_k \\
&= \Omega_{12} \, \mathrm{d}h_1 \wedge \mathrm{d}h_2 + \Omega_{13} \, \mathrm{d}h_1 \wedge \mathrm{d}h_3 + \Omega_{23} \, \mathrm{d}h_2 \wedge \mathrm{d}h_3 \ .
\end{aligned}
\tag{29}
$$

The quantum curvature for a two-level system $\tilde{H} = \sum_{i=1}^{3} h_i \tilde{\sigma}_i$ was computed by Berry [4]: the coefficients are

$$
\Omega_{jk} = \frac{\sum_{i=1}^{3} \epsilon_{ijk} h_i}{2 \left[ \sum_{i=1}^{3} h_i^2 \right]^{3/2}}
\tag{30}
$$

that is

$$
\Omega_{12} = \frac{h_3}{2||h||^{3/2}} \ , \quad \Omega_{13} = -\frac{h_2}{2||h||^{3/2}} \ , \quad \Omega_{23} = \frac{h_1}{2||h||^{3/2}} \ ,
\tag{31}
$$

where $||h|| = \sqrt{h_1^2 + h_2^2 + h_3^2}$.

To express the curvature in terms of the $(X_1, X_2)$ coordinates, note that

$$
\mathrm{d}h_i = \sum_{j=1,2} W_{ij} \mathrm{d}X_j
\tag{32}
$$

so that

$$
\begin{aligned}
\tilde{\Omega} &= \Omega_{12}(W_{11}\mathrm{d}X_1 + W_{12}\mathrm{d}X_2) \wedge (W_{21}\mathrm{d}X_1 + W_{22}\mathrm{d}X_2) \\
&+ \Omega_{13}(W_{11}\mathrm{d}X_1 + W_{12}\mathrm{d}X_2) \wedge (W_{31}\mathrm{d}X_1 + W_{32}\mathrm{d}X_2) \\
&+ \Omega_{23}(W_{21}\mathrm{d}X_1 + W_{22}\mathrm{d}X_2) \wedge (W_{31}\mathrm{d}X_1 + W_{32}\mathrm{d}X_2) \ .
\end{aligned}
\tag{33}
$$

That is

$$
\Omega_n = \Omega_{12}\Theta_3 + \Omega_{13}\Theta_2 + \Omega_{23}\Theta_1
\tag{34}
$$

where

$$
\Theta_1 = W_{21}W_{32} - W_{22}W_{31} \ , \quad \Theta_2 = W_{11}W_{32} - W_{12}W_{31} \ , \quad \Theta_3 = W_{11}W_{22} - W_{12}W_{21} \ .
\tag{35}
$$

We have assumed that $h_1 = h_2 = 0$ at $(X_1, X_2) = (0,0)$. The curvature in the $(X_1, X_2)$ space at $(X, 0)$ is then

$$
\Omega_n(X) = \frac{(\epsilon + W_{31}X)\Theta_3 - W_{21}X\Theta_2 + W_{11}X\Theta_1}{2 \left[ (\epsilon + W_{31}X)^2 + W_{21}^2 X^2 + W_{11}^2 X^2 \right]^{3/2}} \ .
\tag{36}
$$

We now wish to compute the correlation function $C_{nn}(X) = \langle \Omega_n(0)\Omega_n(X) \rangle$ where the expectation value averages over the distributions of the $W_{ij}$ and the $\epsilon$. We shall average over the $W_{i,2}$, then over the $W_{i,1}$, and finally over $\epsilon$. We find the following results for averages over $W_{i2}$:

$$
\langle \Theta_1 \Theta_3 \rangle_{W_{i2}} = -\frac{\sigma_2^2}{2} W_{11}W_{31} \ , \quad \langle \Theta_2 \Theta_3 \rangle_{W_{i2}} = \frac{\sigma_2^2}{2} W_{21}W_{31} \ , \quad \langle \Theta_3^2 \rangle_{W_{i2}} = \frac{\sigma_2^2}{2} [W_{21}^2 + W_{11}^2] \ .
\tag{37}
$$

At this stage it is convenient to change the $W_{i1}$ variables to polar coordinates $(R, \theta, \phi)$

$$
W_{31} = R\cos\theta \ , \quad W_{21} = R\sin\theta\cos\phi \ , \quad W_{11} = R\sin\theta\sin\phi
\tag{38}
$$

so that, noting that the $W_{ij}$ are independent Gaussian distributed variables with zero mean and variance $\sigma_j^2/2$, the probability element for these variables is

$$
\begin{aligned}
\mathrm{d}P & = \frac{1}{(\pi\sigma_1^2)^{3/2}} \exp[-(W_{11}^2 + W_{21}^2 + W_{31}^2)/\sigma_1^2]\mathrm{d}W_{11}\mathrm{d}W_{21}\mathrm{d}W_{31} \\
& = \frac{1}{\pi^{3/2}\sigma_1^3} R^2 \exp(-R^2/\sigma_1^2) \sin\theta \, \mathrm{d}R \, \mathrm{d}\theta \, \mathrm{d}\phi \; .
\end{aligned}
\tag{39}
$$

Now consider the average of $\Omega_n(X)\Omega_n(0)$, evaluated using (36). First we average over $W_{i2}$ using equation (37):

$$
\begin{aligned}
\langle\Omega_n(0)\Omega_n(X)\rangle_{W_{i2}} & = \frac{\sigma_2^2}{8\epsilon^2} \frac{W_{21}^2[(\epsilon + W_{31}X) - W_{31}X]}{[\epsilon^2 + 2\epsilon RX\cos\theta + R^2X^2]^{3/2}} \\
& + \frac{\sigma_2^2}{8\epsilon^2} \frac{W_{11}^2[(\epsilon + W_{31}X) - W_{31}X]}{[\epsilon^2 + 2\epsilon RX\cos\theta + R^2X^2]^{3/2}} \\
& = \frac{\sigma_2^2}{8\epsilon^2} \frac{R^2 \sin^2\theta\epsilon}{[\epsilon^2 + 2\epsilon RX\cos\theta + R^2X^2]^{3/2}} \; .
\end{aligned}
\tag{40}
$$

Now introduce a dimensionless parameter

$$
\lambda = \frac{RX}{\epsilon}
\tag{41}
$$

and compute the average of equation (40) over the $W_{i1}$:

$$
\langle\Omega_n(0)\Omega_n(X)\rangle_{W_{ij}} \sim \frac{1}{4\sqrt{\pi}} \frac{\sigma_2^2}{\epsilon^4\sigma_1^3} \int_0^\infty \mathrm{d}R \, R^4 \exp(-R^2/\sigma_1^2)F(\lambda)
\tag{42}
$$

where

$$
F(\lambda) = \int_0^\pi \mathrm{d}\theta \frac{\sin^3\theta}{[1 + 2\lambda\cos\theta + \lambda^2]^{3/2}} \; .
\tag{43}
$$

Introducing another dimensionless variable

$$
\mu = \frac{\epsilon}{\sigma_1 X}
\tag{44}
$$

we have

$$
\langle\Omega_n(0)\Omega_n(X)\rangle_{W_{ij}} = \frac{1}{4\sqrt{\pi}} \frac{\sigma_2^2\epsilon}{\sigma_1^3 X^5}G(\mu)
\tag{45}
$$

where

$$
G(\mu) = \int_0^\infty \mathrm{d}\lambda \, \lambda^4 \exp(-\mu^2\lambda^2)F(\lambda) \; .
\tag{46}
$$

Finally, we average over $\epsilon$, and multiply by a factor of two because the near-degeneracy can be with either a level above or one below. Hence the contribution to the correlation function from nearly degenerate levels is

$$
\langle\Omega_n(X)\Omega_n(0)\rangle \sim \frac{4\pi^{3/2}}{3} \frac{\rho^3\sigma_1\sigma_2^2}{X} \int_0^\infty \mathrm{d}\mu \, \mu^3 G(\mu) \; .
\tag{47}
$$

This is the dominant contribution to the curvature correlation as $X \to 0$. Evaluating the integrals, we find

$$
F(\lambda) = \int_0^\pi \mathrm{d}\theta \frac{\sin^3\theta}{[1 + 2\lambda\cos\theta + \lambda^2]^{3/2}} = \begin{cases} \frac{4}{3} & 0 < \lambda < 1 \\ \frac{4}{3\lambda^3} & \lambda \geq 1 \end{cases}
\tag{48}
$$

then integration over $\mu$ then $\lambda$ gives

$$C_{nn}(X) \sim \frac{4\pi^{3/2}}{3} \frac{\rho^3 \sigma_1 \sigma_2^2}{X} \ . \tag{49}$$

This is consistent with the expected universal scaling form, equation (22), with

$$f(x) \sim 4\pi^{3/2}/3x \tag{50}$$

as $x \to 0$.

The arguments leading to (49) extend easily to describe the short-range correlations of the curvature of adjacent levels. For small separations both $C_{nn}$ and $C_{n-1,n}$ are dominated by events of near degeneracy of $E_{n-1}$ and $E_n$, but since $\Omega_{n-1}$ and $\Omega_n$ are *anti*correlated during these events, $C_{n-1,n}$ must have the opposite sign, and since $C_{nn}$ receives an independent equal contribution from near degeneracies of $E_n$ and $E_{n+1}$ while $C_{n-1,n}$ does not, the latter should be also be smaller by a factor of two in absolute value. It follows that

$$C_{n-1,n}(X) \sim -\frac{2\pi^{3/2}}{3} \frac{\rho^3 \sigma_1 \sigma_2^2}{X} \ , \tag{51}$$

and therefore $f_1(x) \sim -2\pi^{3/2}/3x$ in the limit as $x \to 0$ (where $f_s(x)$ was defined in equation (24)).

## 3.2 Universality of correlations at arbitrary separations

We investigated the correlation function $C(X)$ numerically for our $M \times M$ GUE random matrix field defined on a unit 2-sphere, as described in subsection 2.5. For this purpose we sampled the joint probability distribution of two matrices $\hat{H}(\mathbf{X}_1)$, $\hat{H}(\mathbf{X}_2)$ subtending angle $\theta$ on the sphere, as well as their $\mathbf{X}$ derivatives. Since different matrix elements of $\hat{H}(\mathbf{X})$ are independent (except for those related by hermiticity), it is sufficient to sample independent realisations of the six-variable joint Gaussian distribution for $H(\mathbf{X}_1)_{jk}, H(\mathbf{X}_2)_{jk} \partial_\alpha H(\mathbf{X}_1)_{jk}, \partial_\beta H(\mathbf{X}_2)_{jk}$ $(\alpha, \beta = 1, 2)$, for each $1 \le j \le k \le M$ to sample a single realisation of $\Omega(\mathbf{X_1})$ and $\Omega(\mathbf{X_2})$. The six-by-six covariance matrix of the matrix elements and their derivatives is straightforwardly determined from the matrix-element correlation function $c(\theta)$ and its derivatives.

This process was repeated for a number $n_\theta$ of equally spaced angular separations between zero (exclusive) and $\theta_m$. The respective values of $n_\theta$ and $\theta_m$ were 120 and $0.18\pi$ for $M = 30$, 120 and $0.15\pi$ for $M = 50$, 100 and $0.1\pi$ for $M = 100$, and 80 and $0.08\pi$ for $M = 150$. The curvature correlation functions reported here were calculated by averaging the product of the curvatures of matrices randomly sampled in this manner. We used $10^6$ realisations of $30 \times 30$ matrices, $5 \times 10^5$ realisations of $50 \times 50$, $10^5$ of $100 \times 100$, and $5 \times 10^4$ realisations of $150 \times 150$ matrices.

Figures 1 and 2 show the numerical results in the form of a data collapse for the scaled diagonal and nearest neighbour correlation functions $f$ and $f_1$ (defined as in equations (22) and (24)) as a function of the scaled separation $x$. Different colours correspond to different choices of $M$, $\tilde{\theta}$, and energy level range. In figure 1 we vary the energy interval of the spectrum, and in figure 2 we show data for two different values of the correlation length $\tilde{\theta}$, combining data for different values of the matrix dimension $M$ in each plot. The quality of the data collapse is a strong indication that the functions $f(x)$ and $f_1(x)$ are universal, and the short distance asymptotics, equations (49) and (51), are confirmed by the matching of the dashed horizontal lines at $4\pi^{3/2}/3$ and $-2\pi^{3/2}/3$ with small $x$ calculations of $xf(x)$ and $xf_1(x)$ (respectively). The solid curves are quadratic-exponential fits

$$xf(x) \approx (4\pi^{3/2}/3)\exp[-(ax + bx^2)] \ , \quad xf_1(x) \approx (-2\pi^{3/2}/3)\exp[-(a_1 x + b_1 x^2)] \ , \tag{52}$$

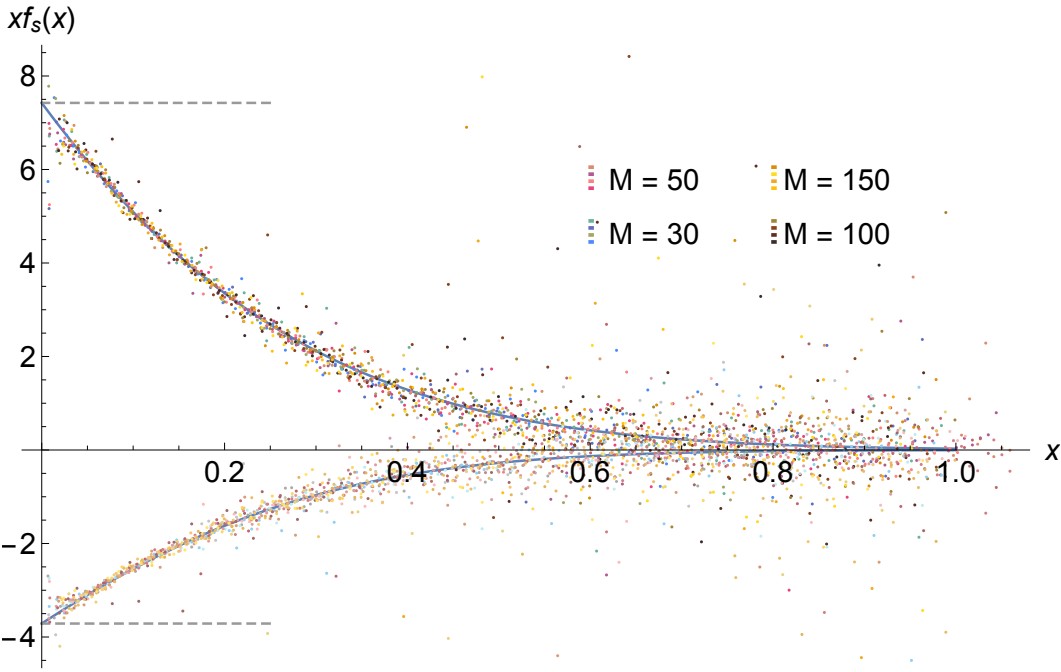

Figure 1: Plot of $xf_s(x)$, obtained by the scaling transformation (24) of the shifted single-level correlation function $C_{n,n+s}(\theta)$, $s = 0, 1$, calculated numerically by Monte-Carlo simulations for the Gaussian random matrix field model with Gaussian matrix element correlation functions and correlation length $\tilde{\theta} = 1$ for several matrix sizes $M$ and energy level groups. Each data set shows the average of $xf_s(x)$ over a range of four ($M = 30$) to twenty ($M = 150$) consecutive energy levels as a function of $x$. Positive (negative) values correspond to $s = 0$ ($s = 1$), respectively, and $s = 1$ data points are shown in lighter hue. The colours next to each value of $M$ represent, from bottom to top, the following energy-level intervals: $16 \leq n \leq 17$, $19 \leq n \leq 20$, $22 \leq n \leq 23$, $25 \leq n \leq 26$, for $M = 30$; $26 \leq n \leq 28$, $30 \leq n \leq 32$, $34 \leq n \leq 36$, $38 \leq n \leq 40$, for $M = 50$; $51 \leq n \leq 55$, $58 \leq n \leq 62$, $65 \leq n \leq 69$, $72 \leq n \leq 76$, for $M = 100$; and $76 \leq n \leq 84$, $89 \leq n \leq 96$, $101 \leq n \leq 108$, $113 \leq n \leq 120$, for $M = 150$. Each energy-level interval is averaged with the corresponding levels below the midpoint of the spectrum.

with $a = 3.56$, $b = 2.03$, $a_1 = 3.43$, $b_1 = 3.55$; we use the fits to estimate the value of integrals which will play a role in section 5:

$$\mathcal{I} = \int_0^\infty \mathrm{d}x \; x \, f(x) \approx 1.69 \; , \quad \mathcal{I}_1 = -2 \int_0^\infty \mathrm{d}x \; x \, f_1(x) \approx 1.58 \; . \tag{53}$$

## 4 Smoothed curvature correlation functions

We present analytical results on the correlation function of the smoothed curvature, $\mathcal{C}(\Delta E, X)$, in the cases where $X = 0$ (subsection 4.1) and $X$ nonzero but small (subsection 4.2), before presenting our numerical results on this correlation function in subsection 4.3. We defined $\bar{\Omega}_\varepsilon(E, \mathbf{X})$ as a local, smoothly weighted average of the $\Omega_n$ in an interval of width $\varepsilon$ centred on $E$, by equation (2), its correlation function $\mathcal{C}(\Delta E, X)$ by (3). We expect the dependence of $\mathcal{C}$ on the energy base point $E_0$ is weak, and only through the mean density of states in the universal part of the smoothed curvature correlation function.

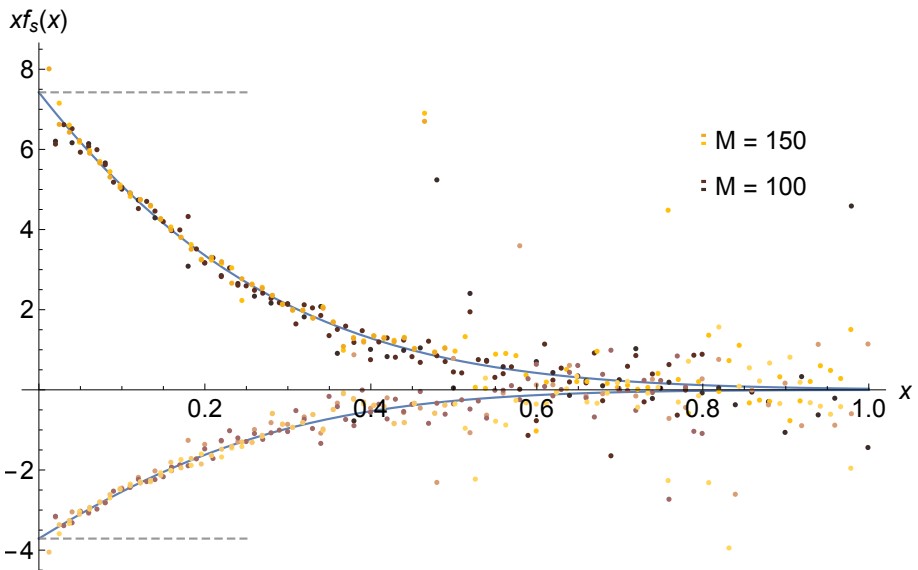

Figure 2: Points show scaled diagonal and nearest neighbour single-level correlation functions, as described in figure 1, except that data are averaged only over the central range of energy levels (as detailed in figure 1), but for different matrix element correlation lengths, confirming universality. The colours next to each value of $M$ represent, from bottom to top, $\tilde{\theta} = 1/2$, 1 ($M = 100$) and $\tilde{\theta} = 1, 2$ ($M = 150$). The dashed lines and solid curves have the same meaning as in figure 1.

## 4.1 One-point correlations

Unlike the single-level curvature correlations, the correlations of the smoothed curvature do not diverge as $\mathbf{X} \to 0$, but degeneracies do play a significant role by causing the $X$-dependence of the correlation function to have a discontinuous derivative. First we consider the correlation function at $X = 0$, before looking at its behaviour for small $X$ in section 4.2.

In this subsection we calculate $\mathcal{C}(\Delta E, 0)$, starting from equation (6). Using equations (2) and (6), and noting that $\Omega_n$ is real, we have

$$\mathcal{C}(\Delta E, 0) = \left\langle \sum_n \sum_{n'} w_\varepsilon(E_0 + \Delta E - E_n) \, w_\varepsilon(E_0 - E_{n'}) \sum_{m \neq n} \sum_{m' \neq n'} \frac{K_{nmn'm'}}{(E_n - E_m)^2 (E_{n'} - E_{m'})^2} \right\rangle \tag{54}$$

where

$$K_{nmkl} = [\partial_1 H_{nm} \partial_2 H_{mn} - \partial_2 H_{nm} \partial_1 H_{mn}][\partial_1 H_{kl} \partial_2 H_{lk} - \partial_2 H_{kl} \partial_1 H_{lk}]^* \, . \tag{55}$$

Now consider how to compute (54) in random matrix theory. Note that $\hat{H}$, $\partial_1 \hat{H}$ and $\partial_2 \hat{H}$ are independent GUE matrices. Because $\hat{H}$ is statistically independent from $\partial_i \hat{H}$, and GUE is invariant under unitary transformations, the matrix elements $\partial_i H_{nm}$ in the eigenbasis of $\hat{H}$ have standard GUE statistics with variances $\sigma_i^2 = \langle |\partial_i H_{nm}|^2 \rangle$. Furthermore, averaging over $\partial_i H_{nm}$ is independent of the average over $\hat{H}$, which is implemented as an average of the eigenvalues, $E_n$. The expectation value of $K_{nmlk}$ for the GUE model is

$$\langle K_{nmkl} \rangle = 2\sigma_1^2 \sigma_2^2 [\delta_{nk} \delta_{ml} - \delta_{nl} \delta_{mk}] \tag{56}$$

so that

$$
\mathcal{C}(\Delta E, 0) = 2\sigma_1^2 \sigma_2^2 \Bigg\langle \sum_n \sum_{n'} w_\varepsilon(E_0 + \Delta E - E_n) w_\varepsilon(E_0 - E_{n'})
$$

$$
\times \sum_{m \neq n} \sum_{m' \neq n'} \frac{\delta_{nn'}\delta_{mm'} - \delta_{nm'}\delta_{mn'}}{(E_n - E_m)^2 (E_{n'} - E_{m'})^2} \Bigg\rangle
$$

$$
= 2\sigma_1^2 \sigma_2^2 \Bigg\langle \sum_n w_\varepsilon(E_0 + \Delta E - E_n) \sum_{m \neq n} [w_\varepsilon(E_0 - E_n) - w_\varepsilon(E_0 - E_m)] \frac{1}{(E_n - E_m)^4} \Bigg\rangle . \quad (57)
$$

The largest terms in the $m$ sum are those with $m$ close to $n$. For such $m$ we can approximate the difference of the window functions by its Taylor series

$$
w_\varepsilon(E_0 - E_n) - w_\varepsilon(E_0 - E_m) = w_\varepsilon'(E_0 - E_n)(E_n - E_m) - \frac{1}{2} w_\varepsilon''(E_0 - E_n)(E_n - E_m)^2 + \cdots \quad (58)
$$

since pairs of terms with $m = n \pm \tilde{m}$ cancel, the sum is dominated by terms of $O(\langle (E_m - E_n)^{-2} \rangle_m$, where $\langle \rangle_m$ stands for averaging over the distribution of $E_m$ with $E_n$ fixed. This expectation value is finite because level repulsion implies that the probability that $|E_m - E_n| < \epsilon$ is $\sim \epsilon$ for $\epsilon$ small. The fast decay of $\langle (E_m - E_n)^{-2} \rangle_m$ as $|m - n|$ increases makes the terms with $m$ close to $n$ dominant, so that the higher order terms in (58) negligible, so that

$$
\mathcal{C}(\Delta E, 0) = -\sigma_1^2 \sigma_2^2 \sum_n \Bigg\langle w_\varepsilon(E_0 + \Delta E - E_n) w_\varepsilon''(E - E_n) S_n \Bigg\rangle \quad (59)
$$

where we define

$$
S_n = \sum_{m \neq n} \Bigg\langle \frac{1}{(E_n - E_m)^2} \Bigg\rangle_m . \quad (60)
$$

Since $S_n$ is dominated by the smallest separations of energy levels, we expect that $S_n \sim A\rho^2(E_n)$ where $A$ is a dimensionless constant. The value of $A$ can be deduced from a 'virial relation' derived by Dyson (see discussion in [6]), who showed that the eigenvalues of a $M \times M$ GUE matrix satisfy

$$
\sum_{n=1}^{M} \sum_{m \neq n} \langle (E_n - E_m)^{-2} \rangle = M(M - 1) . \quad (61)
$$

Combining this with Wigner's semicircle law (27) for the mean density of states we find

$$
S_n \sim \frac{2\pi^2}{3} [\rho(E_n)]^2 . \quad (62)
$$

Hence, in the limit where $\rho\varepsilon \gg 1$

$$
\mathcal{C}(\Delta E, 0) \sim -\frac{2\pi^2}{3} \sigma_1^2 \sigma_2^2 \rho^3 \int_{-\infty}^{\infty} dE \, w_\varepsilon(E + \Delta E) w_\varepsilon''(E)
$$

$$
= \frac{\pi^{3/2}}{6} \frac{\rho^3 \sigma_1^2 \sigma_2^2}{\varepsilon^3} \left[ 1 - \frac{1}{2}\left(\frac{\Delta E}{\varepsilon}\right)^2 \right] \exp\left[ -\frac{\Delta E^2}{4\varepsilon^2} \right] . \quad (63)
$$

Note that this is consistent with the universal scaling form, equation (25), with

$$
g(0, y) = \left[ 1 - \frac{1}{2}\left(\frac{\Delta E}{\varepsilon}\right)^2 \right] \exp\left[ -\frac{\Delta E^2}{4\varepsilon^2} \right] . \quad (64)
$$

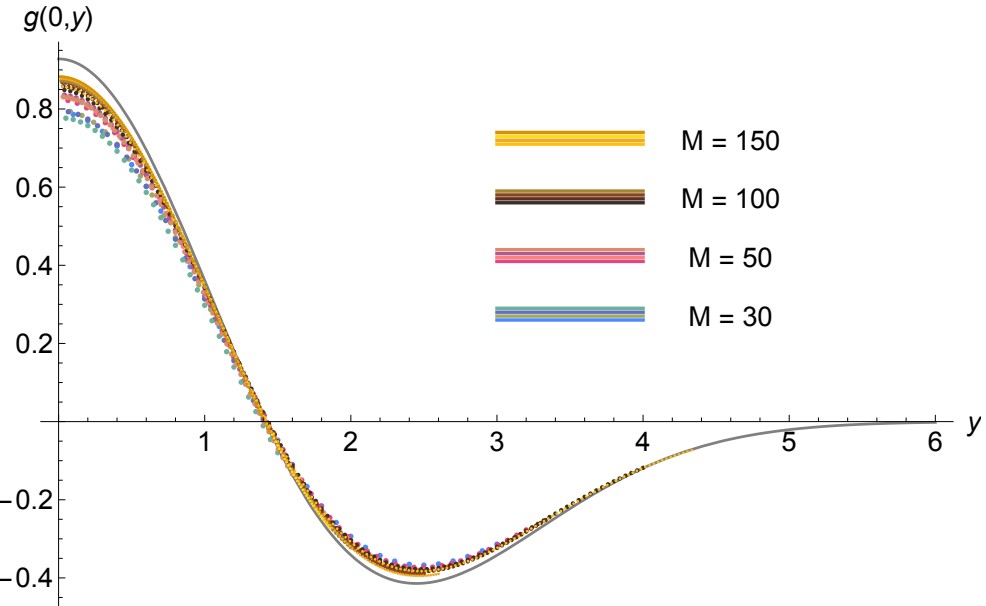

Figure 3: Plot of the numerically calculated (dots) scaled one-point smoothed-curvature correlation function $g(0, y)$, obtained by (25) from $\mathcal{C}(0, \Delta E)$, as a function of $y = \Delta E/\varepsilon$, compared with the exact large-$M$ asymptotic (64) (solid curve). Dots of different colours correspond to different matrix sizes $M$, and several energy window widths $\varepsilon$, all centered at $E_0 = 0$. The colours next to each value of $M$ represent, from bottom top, data for $\varepsilon/\pi = 0.33, 0.42, 0.5, 0.67$ ($M = 30$), $0.31, 0.38, 0.5, 0.63$ ($M = 50$), $0.25, 0.3, 0.4, 0.5$ ($M = 100$), and $0.23, 0.38, 0.54, 0.69$ ($M = 150$).

### 4.2 Two-point correlations at small separations

We can also consider the parameter dependence of the correlation function of the smoothed curvature, namely $\mathcal{C}(\Delta E, X)$, following a similar approach to that leading to equation (49).

The value of $\bar{\Omega}_\varepsilon(E)$ diverges at degeneracies, but $\langle \bar{\Omega}_\varepsilon^2 \rangle$ is finite. The change in the correlation function close to $X = 0$ is determined by nearly-degenerate levels. If $E_n$ is close to $E_{n+1}$, the change in $\Omega_\varepsilon$ due to varying the parameters by a small displacement $(X, 0)$ is

$$\Delta \bar{\Omega}_\varepsilon(X) = w'_\varepsilon(E - E_n) \left( \Delta E(X) \Omega_n(X) - \Delta E(0) \Omega_n(0) \right) \tag{65}$$

where $\Delta E(X) = E_{n+1}(X) - E_n(X)$, and where $\Omega_n(X)$ is given by equation (36), which we write in the form

$$\Omega_n(X) = 4 \frac{(\epsilon + W_{31}X)\Theta_3 - W_{21}X\Theta_2 + W_{11}X\Theta_1}{[\Delta E(X)]^3} \ , \tag{66}$$

where $\Theta_i$ were defined in equation (33), and

$$\Delta E(X) = 2[(\epsilon + W_{31}X)^2 + W_{21}^2 X^2 + W_{11}^2 X^2]^{1/2} \ . \tag{67}$$

We shall consider the quantity $\bar{\Omega}_\varepsilon(E + \Delta E, 0)[\bar{\Omega}_\varepsilon(E, X) - \bar{\Omega}_\varepsilon(E, 0)] \equiv \bar{\Omega}_\varepsilon \Delta \bar{\Omega}_\varepsilon$. This is

$$\bar{\Omega}_\varepsilon \Delta \bar{\Omega}_\varepsilon = w'(E + \Delta E - E_n) w'(E - E_n) \frac{\Theta_3}{\epsilon} \left[ \frac{(\epsilon + W_{31}X)\Theta_3 - W_{21}X\Theta_2 + W_{11}X\Theta_1}{(\epsilon + W_{31}X)^2 + W_{21}^2 X^2 + W_{11}^2 X^2} - \frac{\Theta_3}{\epsilon} \right] .$$
$$\tag{68}$$

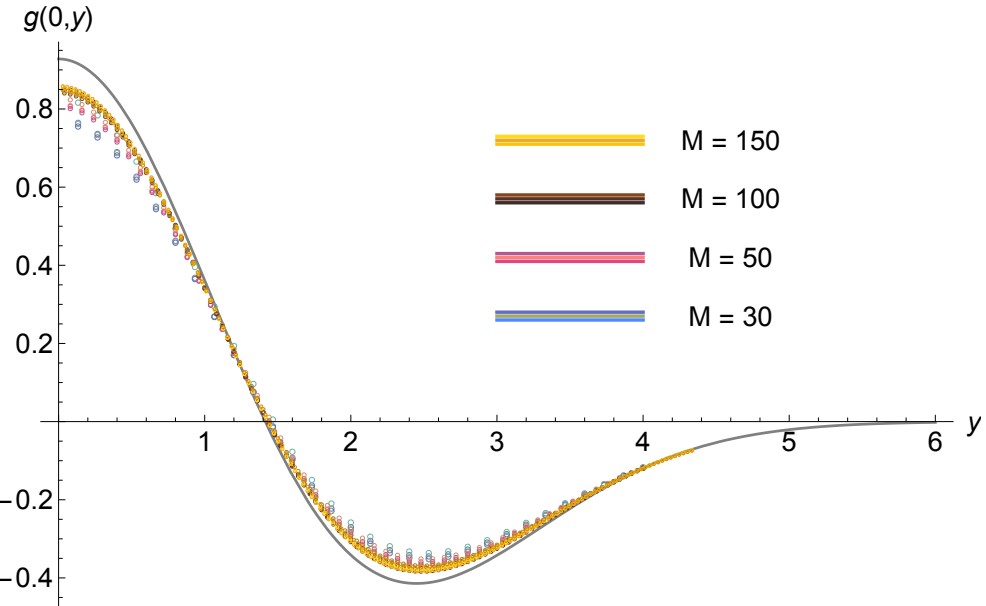

Figure 4:   Same as figure 3 but with correlation functions calculated numerically for energy windows centered at $E_0/\sqrt{M} = 0, 1/2, 1, 3/2$ and fixed width $\varepsilon = \pi/4$.

Taking the expectation value of $\Omega_\varepsilon \Delta\Omega_\varepsilon(X)$, using the same approach and notations as before in section 3, we find

$$\langle \Delta\Omega_\varepsilon(X)\Omega_\varepsilon \rangle_{W_{i2}} = w'_\varepsilon(E + \Delta E - E_n)w'_\varepsilon(E - E_n)\frac{\sigma_2^2}{2}\left[\frac{R^2\sin^2\theta}{\epsilon^2 + 2RX\epsilon\cos\theta + R^2X^2} - \frac{R^2\sin^2\theta}{\epsilon^2}\right]$$

$$\langle \Delta\Omega_\varepsilon(X)\Omega_\varepsilon \rangle_{W_{ij}} = w'_\varepsilon(E + \Delta E - E_n)w'_\varepsilon(E - E_n)\frac{\sigma_2^2}{\sqrt{\pi}\sigma_1^3\epsilon^2}\int_0^\infty dR\ R^4 \exp(-R^2/\sigma_1^2)$$

$$\times \int_0^\pi \sin^3\theta\left[\frac{1}{1 + 2\lambda\cos\theta + \lambda^2} - 1\right]$$

$$\langle \Delta\Omega_\varepsilon(X)\Omega_\varepsilon \rangle = w'_\varepsilon(E + \Delta E - E_n)w'_\varepsilon(E - E_n)\frac{8\pi^{3/2}\rho^3\sigma_2^2\sigma_1^3 X}{3}$$

$$\times \int_0^\infty d\mu\ \mu^5 \int_0^\infty d\lambda\ \lambda^4 \exp(-\lambda^2\mu^2)\mathcal{F}(\lambda) \tag{69}$$

where we have taken expectation values with respect to the $W_{i2}$, then $W_{i1}$ then $\epsilon$ (using the same polar coordinates for the $W_{i1}$, the same definitions of $\lambda$ and $\mu$ as section 3), and

$$\mathcal{F}(\lambda) = \int_0^\pi d\theta\ \sin^3\theta\left[\frac{1}{1 + 2\lambda\cos\theta + \lambda^2} - 1\right]\ . \tag{70}$$

This yields

$$\langle \Delta\Omega_\varepsilon(X)\Omega_\varepsilon \rangle = \frac{8\pi^{3/2}A}{3}[w'_\varepsilon(E - E_n)]^2 \rho^3 \sigma_1^3 \sigma_2^2 X \tag{71}$$

where

$$A = \int_0^\infty d\mu\ \mu^5 \int_0^\infty d\lambda\ \lambda^4 \exp(-\lambda^2\mu^2)\mathcal{F}(\lambda) = -\frac{\pi^2}{8}\ . \tag{72}$$

Finally, we multiply by two, to account for near degeneracies with the level below as well

as the level above, and sum over energy levels. Noting that

$$\sum_n w'_\varepsilon(E + \Delta E - E_n) w'_\varepsilon(E - E_n) \sim \frac{\rho}{2\pi\varepsilon^6} \int_{-\infty}^{\infty} dE \ E^2 \exp\left(-\frac{E^2}{2\varepsilon^2}\right) \exp\left[-\frac{(E + \Delta E)^2}{2\varepsilon^2}\right]$$

$$= \frac{\rho}{4\sqrt{\pi}} \frac{1}{\varepsilon^3} \left[1 - \frac{1}{2}\left(\frac{\Delta E}{\varepsilon}\right)^2\right] \exp\left[-\frac{\Delta E^2}{4\varepsilon^2}\right] \qquad (73)$$

We then have

$$\mathcal{C}(\Delta E, 0) - \mathcal{C}(\Delta E, X) \sim \frac{\pi^3}{12} \frac{\rho^3 \sigma_1^2 \sigma_2^2}{\varepsilon^3} \rho\sigma_1 X \left[1 - \frac{1}{2}\left(\frac{\Delta E}{\varepsilon}\right)^2\right] \exp\left[-\frac{\Delta E^2}{4\varepsilon^2}\right] . \qquad (74)$$

This is consistent with $\mathcal{C}(\Delta E, X)$ having the universal scaling form (25) where the scaling function $g(x, y)$ satisfies

$$g(x, y) \underset{x \ll 1}{\sim} \left(1 - \frac{y^2}{2}\right) \exp(-y^2/4) \left(1 - \frac{\pi^{3/2}}{2}|x| + O(x^2)\right) . \qquad (75)$$

If the $\Omega_n$ were statistically independent, we would expect to find $\mathcal{C} \sim \varepsilon^{-1}$. The fact that $\mathcal{C} \sim \varepsilon^{-3}$ is indicative of cancellation effects due to correlations between the $\Omega_n$, as described by equation (51).

## 4.3 Correlations at arbitrary separations and two-variable universality

We used the data from the Monte-Carlo simulations described in subsection 3.2 to evaluate the smoothed curvature correlation function $\mathcal{C}(\Delta E, X)$ for the parametric GUE model defined in subsection 2.4. We examined the scaling of the correlation function as we varied several paramters: the matrix dimension $M$, the width $\varepsilon$ of the energy interval, the position in the spectrum of the states included in the averaging (which affects the density of states, $\rho$), and the correlation length $\tilde{\theta}$ of the random matrix model.

The scaled numerically calculated single-point correlation function $g(0, y)$ (where $y = \Delta E/\varepsilon$) is shown in figures 3 and 4 overlaid with the large-$M$ exact asympototics (63). The numerical results indeed approach the universal correlations when $M$ increases, but the convergence is slow, with a few percent deviation even for $M = 150$. In figure 3 we vary the width of energy interval, $\varepsilon$, confining the average to states close to the centre of the spectrum. In figure 4 we vary the position of the averaging interval within the spectrum (keeping $\varepsilon = 1/4$ fixed).

The slow convergence as $M$ increases is also observed in figures 5, 6, and 7, where the numerically calculated $g(x, y)$ is plotted as a function of $x = \sigma\rho X$ for a few values of $y = \Delta E/\varepsilon$. All of these figures show data for a wide range of different values of $M$: in figure 5 we vary $\varepsilon$ (keeping close to the centre of the band), in figure 6 we vary the energy interval (keeping $\varepsilon$ fixed), and in figure 7 we compare results for different values of $\tilde{\theta}$. The slow convergence as $M$ increases obscures the scaling collapse of the discontinuity of slope of $g(x, y)$ at $x = 0$. In order to illustrate the validity of (75), the slowly converging part is removed from the correlation function in figures 8, 9, and 10. These show the *subtracted* correlation function $g(x, y) - g(0, y)$, with slopes at $x = 0$ that agree well with the small-$x$ singularity of (75), and exhibiting a very good data collapse confirming the universality of the scaling function $g$.

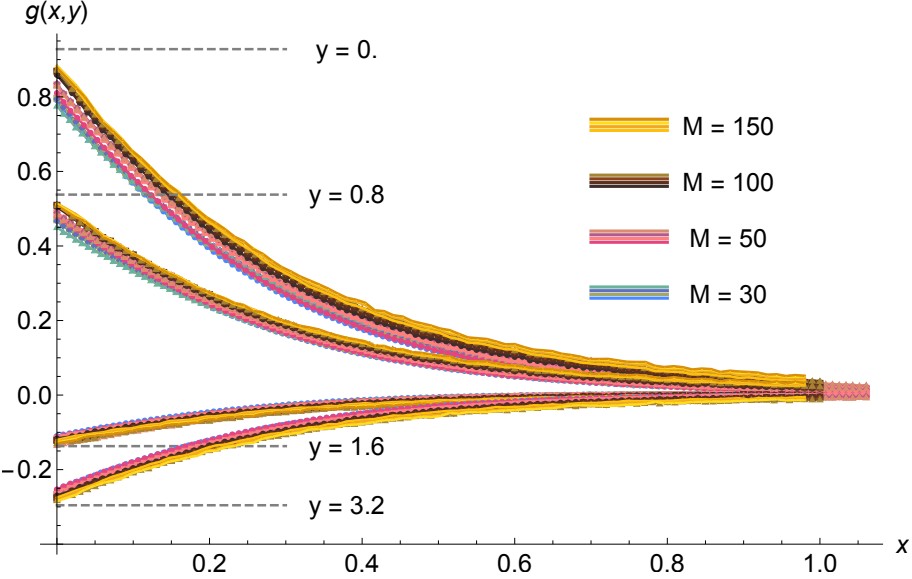

Figure 5: Plot of the numerically calculated scaled smoothed-curvature correlation function $g(x, y)$, obtained by (25) from $\mathcal{C}(X, \Delta E)$, as a function of $x = \rho \sigma X$, for a few fixed values of $y = \Delta E / \varepsilon$. Horizontal dashed lines show the exact large-$M$ asymptotic (64) of $g(0, y)$ for the corresponding $y$, and also serve to label the data sets. Curves of different colours correspond to different matrix sizes $M$, and energy window widths $\varepsilon$, all centered at $E_0 = 0$ with fixed correlation length $\tilde{\theta} = 1$. The gaps at $x = 0$ between the data curves and the dashed lines decrease for larger $M$, as seen in figure 3. Collapse of the data curves confirms two-variable scaling and universality. The colours next to each value of $M$ represent, from bottom to top, data for $\varepsilon / \pi = 0.33, 0.42, 0.5, 0.67$, ($M = 30$), $0.25, 0.38, 0.5, 0.63$ ($M = 50$), $0.3, 0.4, 0.5, 0.6$ ($M = 100$), and $0.38, 0.54, 0.69, 0.85$ ($M = 150$) except that $\varepsilon / \pi = 0.31, 0.38, 0.46, 0.54$ for $y = 1.6, M = 150$, and that for $y = 3.2$, $\varepsilon / \pi = 0.17, 0.25$, ($M = 30$), $0.13, 0.19, 0.25, 0.31$ ($M = 50$), $0.15, 0.2, 0.25, 0.3$ ($M = 100$), and $0.15, 0.23, 0.31$ ($M = 150$).

## 5 Statistics of Chern numbers

Finally, we show how our results on the correlation function of the quantum curvature can be used to make deductions about statistical fluctuations of Chern numbers. The Chern number can be expressed as an integral of the quantum curvature: see equation (4)

First, let us estimate the variance of $N_n$. In our random matrix model it is clear that $\langle N_n \rangle = 0$. We consider the case where the parameter space is isotropic, so that the correlation function $C(X)$ is independent of the direction of $X$. In this case, we write $\sigma_1 = \sigma_2 \equiv \sigma$. Taking the second moment of (4), and using the fact that when $M \gg 1$ the support of the correlation function is small compared to the extent of the parameter space, we have

$$\langle N_n^2 \rangle \sim \frac{1}{2\pi} \mathcal{A} \sigma^2 \rho^2 \mathcal{I} , \quad \mathcal{I} = \int_0^\infty dx \, x \, f(x) \tag{76}$$

where $\mathcal{A}$ is the area of the closed surface of the parameter space, and $f(x)$ is the function defined in equation (23). Numerical evaluation of the integral in (76) (quoted in equation (53)) gives $\mathcal{I} \approx 1.69$. This result is compatible with the results of [21], (based upon data obtained with less powerful computers) which suggest that $\mathcal{I} \approx 1.5$.

We can also use our results to support the hypothesis about correlations of Chern numbers contained in equation (5). We define (by analogy with equation (2)) a smoothed

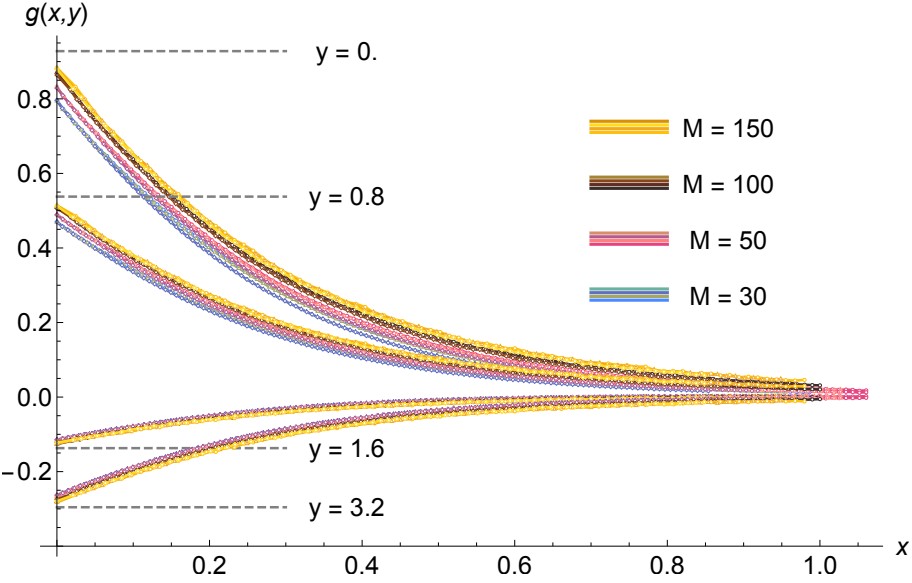

Figure 6: Numerical data and horizontal lines as in 5, except that data are shown for energy windows based at $E_0/\sqrt{M} = 0, 1/2, 1$ and one width for each $M$ and $y$; the energy window width is equal to the second in the list of $\varepsilon$ values shown in figure 5 for the corresponding $M$ and $y$.

Chern number

$$N_\varepsilon(E) = \sum_n N_n w_\varepsilon(E - \bar{E}_n) \tag{77}$$

where $\bar{E}_n$ is an average of $E_n(\mathbf{X})$ over the Brillouin zone. We can express the variance of the smoothed Chern number in two ways. First, express this in terms of the correlation function of $\Omega_\varepsilon(E, \mathbf{X})$:

$$
\begin{aligned}
\langle N_\varepsilon^2 \rangle &= \frac{1}{(2\pi)^2} \int \mathrm{d}\mathbf{X} \int \mathrm{d}\mathbf{X}' \, \langle \Omega_\varepsilon(E, \mathbf{X})\Omega_\varepsilon(E, \mathbf{X}') \rangle \\
&\sim \frac{\mathcal{A}}{(2\pi)^2} \int \mathrm{d}\mathbf{X} \, \mathcal{C}(0, |\mathbf{X}|)
\end{aligned}
\tag{78}
$$

where $\mathcal{A}$ is the area of the Brillouin zone, and in the final step we assume that the correlation is homogeneous, isotropic and short-ranged. The scaling form for the correlation function $\mathcal{C}$, equation (25), indicates that

$$\langle N_\varepsilon^2 \rangle \sim \frac{\sqrt{\pi}\kappa}{12} \frac{\mathcal{A}\rho\sigma^2}{\varepsilon^3} \tag{79}$$

where $\kappa$ is an integral of the scaling function:

$$\kappa = \int_0^\infty \mathrm{d}x \, x \, g(x, 0) \ . \tag{80}$$

Alternatively, we can compute the variance of the smoothed Chern number directly, if we assume that the correlation function of Chern numbers is given by (5). (This hypothesis is equivalent to assuming that the Chern number increments associated with gaps are

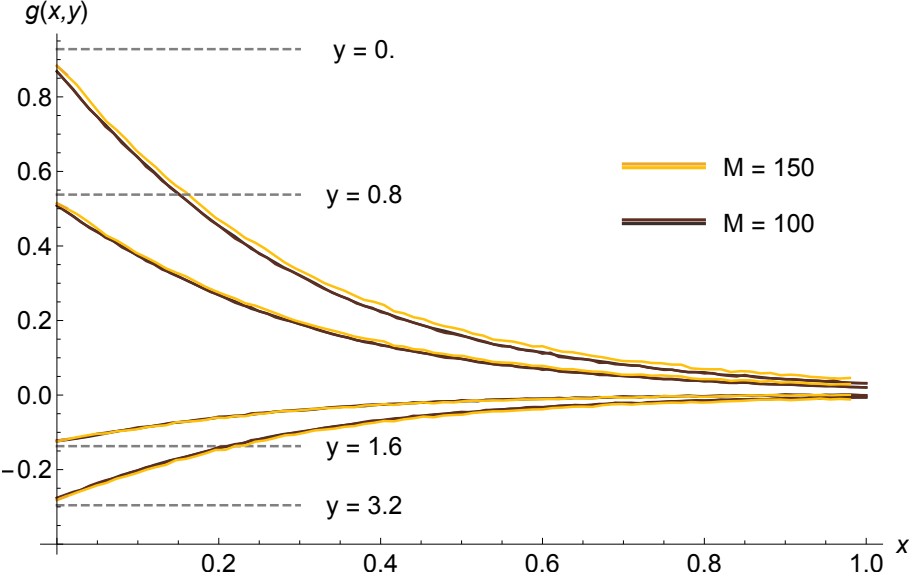

Figure 7: Numerical data and horizontal lines as in 5, except that data are shown for energy single energy window based at $E_0 = 0$, but for different correlation lengths $\tilde{\theta} = 1/2, 1$ ($M = 100$) and $\tilde{\theta} = 1, 2$ ($M = 150$). The energy window widths are equal, respectively for each $M$ and $y$, to the second in the list of $\varepsilon$ values shown in figure 5.

uncorrelated). Using (5) we infer that

$$\langle N_\varepsilon^2 \rangle = \sum_n \sum_m w_\varepsilon(E - E_n) w_\varepsilon(E - E_m) \langle N_n N_m \rangle$$

$$\sim \text{Var}(N_n) \sum_n w_\varepsilon(E - E_n) \left[ w_\varepsilon(E - E_n) - \frac{1}{2} w_\varepsilon(E - E_{n-1}) - \frac{1}{2} w_\varepsilon(E - E_{n+1}) \right] . \tag{81}$$

Expanding the term in square brackets about $E - E_n$, we have:

$$\langle N_\varepsilon^2 \rangle \sim \langle N_n^2 \rangle \sum_n w_\varepsilon(E - E_n) \left[ \frac{1}{2} w_\varepsilon'(E - E_n)(E_{n+1} + E_{n-1} - 2E_n) \right.$$

$$\left. - \frac{1}{4} w_\varepsilon''(E - E_n)[(E_{n_1} - E_n)^2 + (E_{n-1} - E_n)^2] \right] . \tag{82}$$

The terms $E_{n+1} + E_{n-1} - 2E_n$ fluctuate in sign so that the sum containing $w_\varepsilon'(E - E_n)$ as a factor vanishes. The remaining term gives

$$\langle N_\varepsilon^2 \rangle \sim -\frac{\langle N_n^2 \rangle \langle \Delta E^2 \rangle}{2} \rho \int_{-\infty}^{\infty} dE \; w_\varepsilon(E) w_\varepsilon''(E) = \frac{\langle N_n^2 \rangle \langle \Delta E^2 \rangle \rho}{8\sqrt{\pi}\varepsilon^3} \tag{83}$$

where $\langle \Delta E^2 \rangle$ is the mean-squared nearest neighbour spacing. On the basis of the universality hypothesis discussed in section 2, we expect $\langle \Delta E^2 \rangle = \gamma/\rho^2$, where $\gamma$ is a universal dimensionless constant. Using the 'Wigner surmise' distribution for $\Delta E$, equation (8), yields $\gamma = \langle S^2 \rangle = 3\pi/8$ and hence, using (76), we obtain

$$\langle N_\varepsilon^2 \rangle = \frac{3\mathcal{I}}{128\sqrt{\pi}} \frac{\mathcal{A}\rho\sigma^2}{\varepsilon^3} . \tag{84}$$

which is consistent with equation (79). The fact that $\langle N_\varepsilon^2 \rangle$ is proportional to $\varepsilon^{-3}$ is, therefore, an indication that the fluctuations of Chern numbers on successive levels are anticorrelated, as described by equation (5).

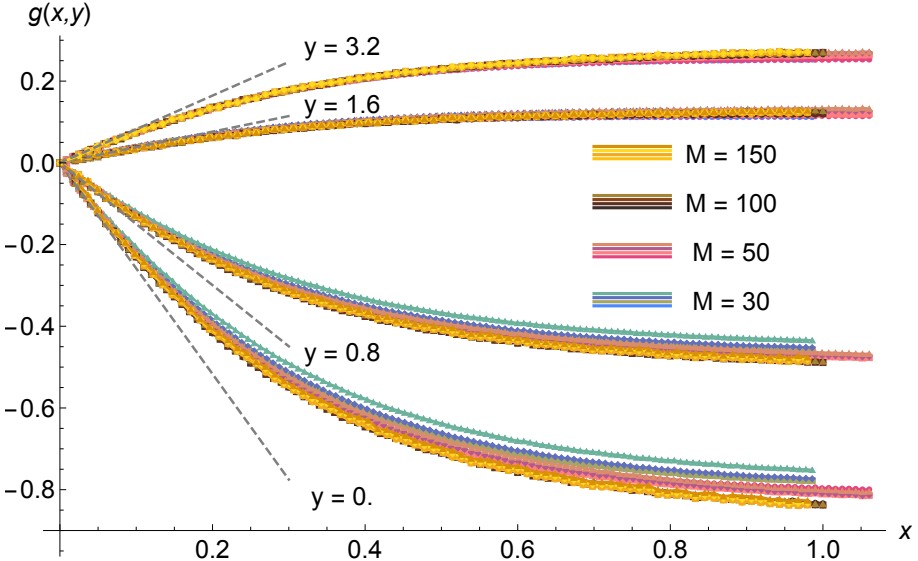

Figure 8: Plot of the same data as in figure 5, showing differences between scaled smoothed curvature correlation function $g(x,y)$ at different points, and the correlation function $g(0,y)$ at the same point, as a function of $x$ for several fixed values of $y$. Straight dashed lines show the small-$x$ asymptotic (75) of $g(x,y)$ for the corresponding $y$, and also serve to label the data sets. Compared to figure 5 the curves exhibits significantly better data collapse, and good agreement with the slopes of the dashed lines.

## 6  Conclusion

We have analysed the universal fluctuations of the adiabatic curvature $\Omega_n$ for complex quantum systems, as exemplified by a parametric GUE model. We find that the correlation function $C(X)$ of $\Omega_n$ has a $X^{-1}$ divergence as $X \to 0$, which is a consequence of near-degeneracies (equations (49), (50)). We also investigated the correlation function numerically, and found that it is consistent with the scaling hypothesis of parametric random matrix theory (equation (22)), as illustrated by figures 3–4.

Because of Landau-Zener transitions these near-degeneracies spread the density matrix over a range of eigenstates, implying that we should also consider a smoothed curvature, $\bar{\Omega}_\varepsilon$. We find that the correlation function $\mathcal{C}$ of $\bar{\Omega}_\varepsilon$ scales as $\varepsilon^{-3}$ (equation (63)), and has a discontinuous first derivative at $X = 0$, described by equations (74) and (75). The numerical evaluation of the smoothed correlation function is illustrated in figures 5–10.

We used these results to analyse the variance of the Chern integers. Their variance is given by (76), which is consistent with the surmise made in [21], and we present evidence that their correlation function is described by (5).

## Acknowledgements

MW is grateful for the generous support of the Racah Institute, who funded a visit to Israel. Both authors are grateful to the Heilbronn Institute and Prof. Jonathan Robbins at the University of Bristol, who organised a workshop where this research was initiated. OG benefitted from helpful discussions with Thomas Guhr and Boris Gutkin.

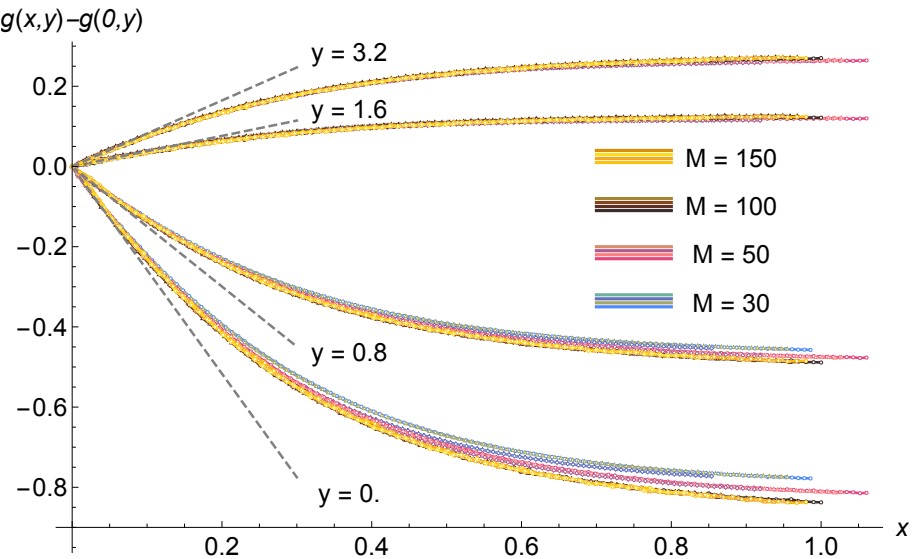

Figure 9: Plot of the same data as in figure 6, subtracted as explained in figure 8. Straight dashed lines have the same significance as in figure 8.

**Funding information**   OG thanks the German-Israeli Foundation for financial support under grant number GIF I-1499-303.7/2019.

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
