# Peer review of "Correlations of quantum curvature and variance of Chern numbers"

_SciPost Physics_

## Round 1 · Referee Report · Pieter W. Claeys (Referee 1) · 2021-4-20

Strengths

1- Presents universality hypothesis for curvature and smoothed curvature correlations. 2- Exact asymptotic results and convincing numeric results. 3- Correlations of the curvature are directly connected to statistical fluctuation of Chern number.

Weaknesses

1- The presentation is slightly confusing at some points.

Report

In this work, the authors investigate the correlation function for the quantum curvature in random matrix models and present a scaling hypothesis for both the curvature and an Gaussian smoothed curvature. The correlation is shown to diverge as the inverse of the (energy) distance at short distances, whereas the smoothed correlations are finite but singular in this limit. Exact asymptotics in this limit are obtained by modelling the curvature using a two-level subspace, for which the curvature can be exactly calculated, returning the expected asymptotics after an appropriate averaging over the matrix elements. These asymptotics are supported by Monte Carlo simulations, which are then also used to calculate the scaling function at longer energy distances, supporting a more general scaling hypothesis. Finally, the variance on Chern numbers is expressed in terms of the calculated correlation functions.

These results are sound and interesting, both the analytic and numeric part, and presented in a convincing way, such that I am happy to recommend this paper for publication. I have some minor comments that the authors may choose to address, but this will not determine my opinion of the paper.

Requested changes

As mentioned in the report, none of these are requested changes, but are only meant to possibly clarify some parts of the manuscript. 1- Can the authors comment on why the spread on the scaling collapse, as illustrated in Figures 1 and 2, increases as $x$ increases? 2- It might be good to comment on what features are expected to be present in correlations of the curvature of the eigenstates of general Hamiltonians, and if there are specific restrictions on energy scales, the parameters $X$ and the Hamiltonian, etc. that need to be met in order to observe the presented scaling. 3- The scaling of the correlation function is currently presented in Section 2.4, but both the preceding section 2.3 and the following Section 3.1 (finishing the derivation started in 2.3) deal only with the small-separation asymptotics, which might give the impression the universality hypothesis depends on these small separations. It might be useful to move Section 2.4 closer to the beginning of the paper or already present these scaling functions in the introduction in order to make these results stand out from the surrounding derivation. 4- After Eq. (3), it might be useful to mention that the angle brackets are expectation values w.r.t. the random matrix ensemble, in order to avoid confusion. 5- There are some typos in the manuscript: $\rho$ is not defined when first introduced after Eq. (2); "the use random matrix models"; In Eq. (5) I assume a $N_n$ should be $N_m$; "unitarily equivalent Hamiltonian [20]"; There is a bracket missing in $O(\dots)$ after Eq. (58); "higher order terms in (58) negligible", "paramters".

  • validity: high
  • significance: high
  • originality: high
  • clarity: good
  • formatting: good
  • grammar: perfect

Author:  Michael Wilkinson  on 2021-05-06  [id 1412]

(in reply to Report 1 by Pieter W. Claeys on 2021-04-20)

We are grateful to the referees for their very generous assessment and constructive criticisms.

In response to the report from Pieter Claeys, we have revised the Introduction to be more explicit about the
type of expectation value which is considered. Also, at the end of section 2.4 we emphasise that the scaling relations are expected to be applicable when the Hamiltonian varies sufficiently slowly. We decided against moving section 2.4 forward because it depends upon definitions developed in earlier sections.

The greater fluctuations for larger x in figures 1 and 2 are at least partially explained by the fact that we’re showing xf(x) so that if the fluctuations in f(x) are independent of x, we should expect the fluctuations in xf(x) to grow with x.

We corrected the typos mentioned in the report.

A revised manuscript has been uploaded to arXiv.

---

## Round 1 · Referee Report · Anonymous (Referee 2) · 2021-4-25

Strengths

  • The manuscript identifies approximate degeneracy points as the dominant source of Berry curvature correlations and demonstrates the universal Berry curvature statistics
  • The conclusions are well-founded and fully supported by the numerics

Weaknesses

  • After reading the manuscript I cannot confirm with certainty that the RMT model is sufficiently substantiated, and that the findings apply to realistic systems.

Report

My background

I am familiar with the formalism, and can evaluate the relevance of the research question and correctness of the approach, however I don't have a sufficiently close familiarity with RMT to be certain that the authors cite all the relevant references.

Background and novelty

The approach critically relies on some of the earlier findings, some by the second author. On a technical level, the approach seems relatively standard, which is appropriate for the research question. To the best of my understanding the main conclusions of the manuscript were not previously known.

Correctness and reproducibility

  • The calculations in the manuscript are clearly supported by numerics, and at a glance, the numerical simulations are correct.
  • Some of the central assumptions of the model are not sufficiently substantiated or explained. Specifically, the authors state that considering $⟨Ω⟩ = 0$ is sufficiently universal, and that it is sufficient to only consider local correlations of $H(X)$. In realistic systems (e.g. band structures of supercells), neither assumption would hold, and therefore both assumptions need to be clearly substantiated even though they make intuitive sense.
  • I believe that the readers would benefit if the code for generating the data, and for producing the plots was uploaded to a data repository (zenodo/datadryad/figshare).

Relevance

The work may have applications to quasiclassical dynamics in chaotic systems, or to RMT studies of other topological features like statistics of Weyl points. While both directions are, to the best of my understanding, not very active, I believe there are sufficiently many researchers who would be interested to learn about the findings of the manuscript.

Clarity

The manuscript is written in a systematic and clear fashion, and it is accessible to non-experts. While some parts are more technically involved, I believe the authors did an overall good job. That said, I think the manuscript would benefit from an expanded motivation of the RMT setting. Perhaps the authors could even study a different ensemble and demonstrate that the Berry curvature statistics agrees with the RMT predictions.

Based on the above I believe the manuscript satisfies the requirements of SciPost Physics.

Requested changes

The main change I would like to request is publishing code in a specialized data repository, as required by the journal rules.

Other than that I suggest the authors to consider clearly motivating the assumptions of zero Berry curvature and only local Hamiltonian correlations. I think the manuscript would also benefit from a confirmation that the RMT predictions are valid in an ensemble with more structure. I do not consider these changes a requirement for the manuscript to be suitable for publication.

  • validity: high
  • significance: high
  • originality: top
  • clarity: high
  • formatting: perfect
  • grammar: perfect

Author:  Michael Wilkinson  on 2021-05-20  [id 1440]

(in reply to Report 2 on 2021-04-25)
Category:
remark

In response to the report from the anonymous referee, we have revised both the Introduction and the Discussion
sections our text to emphasise that the use of a model with zero mean curvature is intended to capture short-ranged fluctuations of the curvature.

Our model depends on two nonuniversal parameters, \rho and \sigma, and we expect local universality in regions in energy and parameter space that are small enough that these are approximately constant.

We intend to upload the code and data to one of the repositories mentioned by this referee promptly .

---

## Editorial Decision

resubmitted